Improved salt tolerance of Chenopodium quinoa Willd. contributed by Pseudomonas sp. strain M30-35

Cai Deyu 1 2 3
Xu Ying 2
Zhao Fei 1
Zhang Yan 2
Duan Huirong duanhuirong@caas.cn 4
Guo Xiaonong gxnwww@xbmu.edu.cn 1 2 3
1 Key Laboratory of Biotechnology and Bioengineering of State Ethnic Affairs Commission, Biomedical Research Center, Northwest Minzu University , Lanzhou , China
2 College of Life Science and Engineering, Northwest Minzu University , Lanzhou , China
3 China-Malaysia National Joint Laboratory, Biomedical Research Center, Northwest Minzu University , Lanzhou , China
4 Lanzhou Institute of Husbandry and Pharmaceutical Sciences, Chinese Academy of Agricultural Sciences , Lanzhou , China
Amanullah Amanullah
Electronic publication date: 2021 Jan 13
Publication date: 2021
Volume: 9
Electronic Location ID: e10702
Received 2020 Sep 8; Accepted 2020 Dec 14
Copyright: ©2021 Cai et al.
Copyright year: 2021
Copyright holder: Cai et al.
License: This is an open access article distributed under the terms of the Creative Commons Attribution License, which permits unrestricted use, distribution, reproduction and adaptation in any medium and for any purpose provided that it is properly attributed. For attribution, the original author(s), title, publication source (PeerJ) and either DOI or URL of the article must be cited.
License URL: https://creativecommons.org/licenses/by/4.0/

Keywords: Pseudomonas sp. strain M30-35, Chenopodium quinoa, Salt stress, Photosynthesis, Plant-growth-promoting

Funding: National Natural Science Foundation of China 31760242 Northwest Minzu University 1001070204 11080306 Ministry of Education of China for an Innovative Research Team in University IRT 17R88 Fundamental Research Funds for the Central Universities 31920190021 This research was supported by the National Natural Science Foundation of China (31760242), the Characteristic discipline of bioengineering construction for the special guide project of the “world-class universities and world-class disciplines” of Northwest Minzu University (1001070204, 11080306), the Ministry of Education of China for an Innovative Research Team in University (IRT 17R88) and the Fundamental Research Funds for the Central Universities grant number (31920190021). The funders had no role in study design, data collection and analysis, decision to publish, or preparation of the manuscript.

==============================
Background

Plant-growth-promoting rhizobacteria (PGPR) can promote plant growth and enhance plant tolerance to salt stress. Pseudomonas sp. strain M30-35 might confer abiotic stress tolerance to its host plants. We evaluated the effects of M30-35 inoculation on the growth and metabolite accumulation of Chenopodium quinoa Willd. during salt stress growth conditions.

Methods

The effects of M30-35 on the growth of C. quinoa seedlings were tested under salt stress. Seedling growth parameters measured included chlorophyll content, root activity, levels of plant- phosphorus (P), and saponin content.

Results

M30-35 increased biomass production and root activity compared to non-inoculated plants fertilized with rhizobia and plants grown under severe salt stress conditions. The photosynthetic pigment content of chlorophyll a and b were higher in M30-35-inoculated C. quinoa seedlings under high salt stress conditions compared to non-inoculated seedlings. The stability of P content was also maintained. The content of saponin, an important secondary metabolite in C. quinoa, was increased by the inoculation of M30-35 under 300 mM NaCl conditions.

Conclusion

Inoculation of M30-35 rescues the growth diminution of C. quinoa seedlings under salt stress.

Introduction

Salinity is a major threat to crop production, especially in arid and semi-arid areas (Flowers & Colmer, 2008; Schleiff, 2008). Salt stress has a negative effect on plant growth, and development (Yu et al., 2018), and it is challenging to determine the efficient use of saline soilsfor food production. Two primary strategies, modified agricultural practices and the development of new varieties adapted to saline soils, are being used for growing crops in saline soils. These strategies have produced useful, results but they are still in the early stages of development (Panta et al., 2014). The use of plant-growth-promoting rhizobacteria (PGPR) may help enhance plant stress resistance (Elesawi et al., 2018; Gururani & Upadhyaya, 2013; Kuzyakov & Razavi, 2019).

PGPR may improve safe crop-management practices and stress resistance (Tiwari & Lata, 2018). Kloepper & Schroth (1978) proposed the concept of PGPR, which can promote plant growth and crop yield and improve plant resistance to biotic or abiotic stress. PGPR are also important in reducing oxidative damage and the negative effects of abiotic stresses such as drought, extreme temperatures and heavy metal stress (Bresson et al., 2013; Gururani & Upadhyaya, 2013; Khan et al., 2019; Vardharajula et al., 2011). Pseudomonas sp. M30-35 is a novel PGPR strain isolated from the rhizosphere of Haloxylon ammodendron (C.A.Mey.) Bunge (Chenopodiaceae) in the Tengger desert of northern China. M30-35 inoculation significantly improved the growth performance of perennial ryegrass (Lolium perenne L.) under high salt stress (He et al., 2018). Biofertilizer containing M30-35 improved the efficacy of biofertilizer alone (Gou et al., 2020). It is not known if M30-35 is associated with protective beneficial effects on halophytes. Hence, further studies are needed.

Chenopodium quinoa is an annual halophyte that originated in the Andes Mountains of South America and has been cultivated for more than 7,000 years (Jacobsen, Liu & Jensen, 2009). C. quinoa grows well under moderate salt stress (100–200 mM of NaCl), and can withstand a concentration of 400 mM of NaCl (Jacobsen, Mujica & Jensen, 2003). C. quinoa is rich in proteins, amino acids, dietary fibers, vitamins, minerals, and beneficial phytochemicals such as saponins (Navruz-Varli & Sanlier, 2016). Thus, C. quinoa is an ideal plant for evaluating the effects of PGPR on plant salt tolerance. However, the roles of PGPR on the salt resistance modulation of C. quinoa may be limited.

In the present study, the effects of M30-35 on the growth and salt tolerance of C. quinoa were analyzed under different salt treatments, and the physiological responses were compared.

Materials & Methods

Bacterial strain culture

Luria broth (LB) (5 g/L yeast extract (Solarbio, Beijing, China), 10 g/L peptone (Solarbio, Beijing, China), 10 g/L sodium chloride (Guangfu, Tianjin, China), PH 7.2) were used for bacterial culture. Pseudomonas sp. strain M30-35 (obtained from the State Key Laboratory of Grassland Agro-Ecosystems, College of Pastoral Agriculture Science and Technology, Lanzhou University, Lanzhou, China; the strain is deposited in the Marine Culture Collection of China with preservation number 1k03247.) was incubated, in the dark, at 28 °C, with shaking at 220 rpm (YiHeng, THZ-98AB, ShangHai, China), for 24 h. Bacteria were harvested at an OD600 between 0.9 and 1.0, then diluted to 109 colony forming units (CFU) mL−1, as measured by optical density and serial dilutions (Zhao et al., 2016).

Plant growth and treatments

The C. quinoa seeds (Animal Husbandry, Pasture and Green Agriculture Institute, GAAS, Gansu, China) were surface sterilized with 0.5% potassium permanganate solution for 10 min and washed with sterile water until colorless (Zheng et al., 2017). The seeds were immersed in LB liquid medium (blank control) and Pseudomonas sp. M30-35 bacterial solution (treatment group) for 20 min. Then, the C. quinoa seeds were germinated on moist filter paper in Petri dishes and kept for 48 h at 25 °C in darkness (Gururani & Upadhyaya, 2013). After germination, the plantlets were transferred to plastic pots (upper diameter, 48 mm; bottom, 23 mm; depth, 50 mm). The pots contained vermiculite, which was previously heated at 121 °C for 20 min for sterilization. Each pot was inoculated with 1 mL of M30-35 suspension culture or 1 mL of liquid LB medium. Each treatment was irrigated with half-strength Hoagland solution (2 mM KNO3, 0.5 mM NH4H2PO4, 0.5 mM Ca(NO3)2, 0.5 mM MgSO4, 0.5 mM Fe-citrate, 92 µM H3BO3, 18 µM MnCl2 ⋅ 4H2O, 1.6 µM ZnSO4 ⋅ 7H2O, 0.6 µM CuSO4 ⋅ 5H2O, and 0.7 µM (NH4)6Mo7O24 ⋅ 4H2O) (2 L). The plants were grown in a greenhouse with a 16:8 h (L:D) photoperiod, 26 °C day and 22 °C night temperature, and a relative humidity of 65 ± 10% (He et al., 2018).

Plants aged 14 d were subjected to salt stress by treatment with 0, 150, or 300 mM of NaCl solutions. Treated seedlings were harvested at 7 d and 21 d after salt treatment to record the plant biomass and physiological measurements (Su et al., 2017).

Determination of the seedling biomass

Plants were removed from the pots, and the roots were washed with sterilized water to remove attached soil. Then, the seedlings were divided into shoot and root portions. The samples were dried in an oven at 70 °C for 4 d, and the dry weight was measured (Elesawi et al., 2018).

Determination of chlorophyll contents

The relative chlorophyll content was measured using the ethanol-acetone method, according to the method previously reported (Han et al., 2014). Chlorophyll was extracted from the third compound leaf (50 mg) with an 80% (v/v) cold acetone and 95% ethyl alcohol mix (1:1). A spectrophotometer (Shimadzu, UV-1800, Suzhou, China) was used to read the absorbance at 663 nm for chlorophyll a and 645 nm for chlorophyll b contents. Six samples were measured in each treatment. Each measurement was conducted at 10:00 a.m.

Determination of root activity

Root activity was determined using the triphenyltetrazolium chloride (TTC) colorimetric method (Muhammad et al., 2019). C. quinoa seedling roots were washed, and the root tip samples were excised. Then, the root tips were dried with filter paper. The reaction mixture consisted of 0.2 g root samples, 5 mL of PBS (pH 7.0), and 5 mL of 0.4% TTC in a tube, with the root tips fully immersed in the solution for 2 h at 37 °C. A 2 mL amount of 1 M of sulfuric acid was added to stop the reaction. The roots were removed from the reaction solution, blotted, and homogenized using a mortar and pestle. The red extraction was collected by ethyl acetate to determine the final volume of 10 mL. The absorbance was determined at 485 nm against a blank of ethyl acetate. The standard curve reduction value was TTC(y): y = 0.0021 ×  − 0.0113, R = 0.9945, where y represents the absorbance reading. The root vigor = TTC(x)/W × time reduction value, where W is the root fresh weight, and the time was 2 h.

Determination of total P content

Phosphorus content was determined by the molybdenum blue spectrophotometry method (Tisarum et al., 2020). Dried shoot and root samples (100 mg) were added to 8 mL of H2SO4 and soaked overnight. Then, this was digested at 300 °C for 90 min until fumes were produced. After slight cooling, 0.5 mL of H2O2 was added to the tubes, and heated at 300 °C until the solution was colorless or clear. The volume of extract was maintained by adding distilled water up to 100  mL in the volumetric flask. Then, 10 mL of the previous digestion liquid was transferred to a 50 mL volumetric flask, and dinitrophenol indicator was added. Afterwards, 5 mL of molybdenum antimony anti-coloring agent was added, shaken well, and added with up to 50  mL of ultrapure water. Total P (mg g−1 DW) was measured at 420 nm by a spectrophotometer using KH2PO4 as the calibration standard.

Saponin quantification

The content of total saponin in the different salt treatments of C. quinoa was determined by colorimetry (Liao et al., 2017) using oleanolic acid as the standard. C. quinoa saponins were retrieved by microwave heating. The dried shoot and root samples (500 mg) were extracted with 10 mL of 75% alcohol placed in a ultrasonic bath for 30 min at 60 °C. The extract solution was then evaporated to dryness in 60 °C water. Next, both the 200 µL 5% vanillin acetic acid (vanillin: glacial acetic acid was 5:100, W/V) and 800 µL perchloric acid were added to different test tubes. The liquid was boiled for 15 min, and up to 5 mL of glacial acetic acid was added in a volumetric flask after cooling. The absorbance was determined at 546 nm using a spectrophotometer against the reagent blank.

Data analysis

Excel 2010 was used to collect and process the original data. All data were analyzed by one-way analysis of variance (ANOVA). Duncan’s post-hoc multiple comparison tests were used to identify differences between means at a significance level of P ≤ 0.05 using SPSS version 21.0 (IBM Corporation) statistical software. Results of the growth and physiological data are presented as means ± standard error. Data were plotted using the ORIGIN software.

Results

M30-35 effects on C. quinoagrowth

With increased salinity for 7 d and 21 d, the dry weight of shoots decreased in the control and the M30-35 medium. For the 7 d treatment, different NaCl concentrations had no effect on the shoot dry weight of the control, while 150 mM of NaCl increased the shoot dry weight of seedlings inoculated with M30-35 (Fig. 1A). For the 21 d treatment, M30-35 and the control had similar levels of reduced growth. In contrast, M30-35 reduced the decline of shoot dry weight (Figs. 1A and 1B). M30-35 slightly improved shoot growth, compared to the control, under NaCl treatment.

Figure 1 Effect of M30-35 on the growth of C. quinoa under salt treatments (0, 150 and 300 mM of NaCl).

(A) The shoot dry weight of quinoa after 7 d; (B) the shoot dry weight of C. quinoa after 21d; (C) the NaCl treatment for 7 d, and root dry weight; (D) the NaCl treatment for 21d, and root dry weight. The values are the means, and the bars are standard errors (n = 6). Columns with different letters indicate significant differences among treatments at P < 0.05 (ANOVA and Duncan’s multiple comparison test); an asterisk (*) refers to the significant difference between the LB and M30-35 mediums (P < 0.05).

We compared the effect of adding Luria broth (LB) to plants grown in the presence or absence of M30-35. The roots maintained normal growth after 7 d in the NaCl treatment. However, roots were reduced by exposure to NaCl for 21 d. Compared to the control, M30-35 significantly improved the dry weight of roots by 56.67% and 51.97% at 0 and 150 mM NaCl treatments for 7 d. After 21 d, M30-35 provided a non-significant improvement in the dry weight of roots under salt treatment. The seedlings inoculated with M30-35 had increased root dry weights under different NaCl treatments (Figs. 1C and 1D).

Influence of M30-35 on the chlorophyll content of C. quinoa

After a prolonged time period, both chlorophyll a and chlorophyll b in C. quinoa seedlings showed similar changes under the NaCl treatments (Fig. 2). After NaCl treatment for 7d, the chlorophyll a and chlorophyll b levels in the controls significantly decreased with the increase in salt concentrations. However, the levels of chlorophyll a and b in the M30-35 treatment maintained relative stability (Figs. 2A and 2C). Under 300 mM NaCl conditions the chlorophyll a content of the M30-35 treatment significantly increased by 40.48%, compared to the controls (P < 0.05). In contrast, the content of chlorophyll a and b in the control or M30-35 medium in the NaCl treatment at 21 d remained stable except for a significant increase in chlorophyll b (66.54%) under the 300 mM NaCl treatment (Figs. 2B and 2D).

Figure 2 Effects of M30-35 on the leaf chlorophyll a content (A and B) and leaf chlorophyll b content (C and D) of C. quinoa under salt stress (0, 150, and 300 mM of NaCl).

(A and C) NaCl treatments for 7 d. (B and D) NaCl treatments for 1 d. The values are means, and the bars are standard errors (n = 6). Columns with different letters indicate the significant differences among treatments at P < 0.05 (ANOVA and Duncan’s multiple comparison test); an asterisk (*) refers to the significant difference between the LB and M30-35 mediums (P < 0.05). Two asterisks (**) refer to the difference between the LB and M30-35 mediums (P < 0.01).

Impact of M30-35 on the root activity of C. quinoa

The activity of root system is related to the survival rate of plant, which is an important index to reflect the quality of plant seedlings and evaluate the response of plants to stress (Zhang et al., 2019). With increasing NaCl concentration, the C. quinoa root activity in both the control and M30-35 medium improved at 150 mM, but subsequently declined by 300 mM, at both 7 d and 21 d (Fig. 3). Compared to the control group, the root activity of seedlings inoculated with M30-35 improved by 11.88% (P < 0.05) at 150 mM NaCl for 7 d, and improved by 27.12% (P < 0.05) under 300 mM of NaCl for 21 d.

Figure 3 Effects of M30-35 on the root activity of C. quinoa under salt treatment (0, 150, and 300 mM of NaCl).

(A) The root activity of plants measured after 7 d; (B) the root activity of plants measured after 1 d. The values are the means and the bars indicate the SEs (n = 3). Columns with different letters indicate significant differences among treatments at P < 0.05 (ANOVA and Duncan’s multiple comparison test). An asterisk (*) refers to the significant difference between the LB and M30-35 mediums (P < 0.05).

The effect of M30-35 on the total P contents of C. quinoa

The P content of all LB-treated plants were reduced after exposure to salt stress (Fig. 4). The inoculation of M30-35 maintained a relatively stable P content in plants under salt stress. The P content of the M30-35 treatment was significantly higher than that in the controls under 150 mM of NaCl for 7 d (Fig. 4A).

Figure 4 Comparison of the P content in plants under salt treatments (0, 150, and 300 mM of NaCl).

(A) The total P content of plants measured after 7 d; (B) the total P content of plants measured after 1 d. The values are the means and the bars indicate the SEs (n = 3). Columns with different letters indicate a significant difference among treatments at P < 0.05 (ANOVA and Duncan’s multiple comparison test). An asterisk (*) refers to the significant difference between the LB and M30-35 mediums (P < 0.05).

Effect of M30-35 on the saponin content in C. quinoa seedlings

Similar to the change pattern of root activity, the total saponin content initially increased and then decreased. In the 300 mM NaCl treatment, M30-35 significantly increased the saponin content by 17.40% (P < 0.05). However, M30-35 had no significant effect on the saponin content in the 150 mM NaCl treatment (Fig. 5).

Figure 5 Effects of M30-35 on the content of saponins in C. quinoa under salt treatment (0, 150, and 300 mM of NaCl).

The values are means and the bars indicate the SEs (n = 3). Columns with different letters indicate a significant differences among treatments at P < 0.05 (ANOVA and Duncan’s multiple comparison test). An asterisk (*) refers to the significant difference between the LB and M30-35 mediums (P < 0.05).

Discussion

PGPR can improve the salt tolerance of plants and also promote their growth and development (Safdarian et al., 2019). Pseudomonas includes many common species that can colonize plant roots (Han et al., 2014). These strains can have significant interactions with host plants and induce plant growth (Egamberdieva et al., 2017; Han et al., 2014). Under salt stress, PGPR can increase the root length, root surface area, and number of root tips, thereby enhancing the uptake of plant nutrition, and ultimately improving plant growth and development under stress (Egamberdieva & Kucharova, 2009; Trivedi et al., 2020). Roots colonized by the M30-35 strain enhanced the tolerance of the whole plant to salt stress, and the enhancement was reflected by increasing plant root growth and nutritional status (He et al., 2018). Halotolerant strains of bacteria (Enterobacter sp. [MN17] and Bacillus sp. [MN54]) can promote the plant health and performance of C. quinoa (Yang et al., 2016). The present results also demonstrated that M30-35 significantly increased the dry weight of roots under saline conditions and also increased the dry weight of the shoots. Serratia liquefaciens KM4 promoted maize (Zea mays L.) salt tolerance by inducing the accumulation of the biomass yield (Elesawi et al., 2018). M30-35 might improve the salt tolerance of C. quinoa in a similar manner. However, this hypothesis requires further verification.

Photosynthesis is the basis for increasing in plant biomass and provides the energy needed for metabolism. Changes in chlorophyll levels can be considered to be a biochemical marker of salt tolerance, which reflects the physiological state of a plant’s response to salt (Bernal-Vicente et al., 2018; Singh & Gautam, 2013). Therefore, chlorophyll plays key roles in plant development and stress resistance processes, and its concentration has been shown to decline less rapidly in strong stress-resistant plants, when compared to stress-sensitive plants (Liu & Jiang, 2010). In the present study, the application of M30-35 reduced the decline of chlorophyll a and b in the leaves of C. quinoa seedlings after 7 d of NaCl treatment, indicating the potential role of M30-35 in protection against salt stress. In the 300 mM NaCl treatment, seedlings inoculated with M30-35 had significantly increased chlorophyll a content after the 7 d treatment, and chlorophyll b content in the 7 d treatment, and increased chlorophyll b levels after the 21 d treatment. These results suggest that M30-35 could maintain the C. quinoa chlorophyll content, especially when seedlings were under severe salt stress. These results were similar to observations made on white clover and ryegrass (Han et al., 2014; He et al., 2018).

The plant root system is among the most important components of plants, and is sensitive to salt stress (Liu et al., 2019). Salt stress can reduce root vigor which was defined as the capacity for deoxidization, and affect absorption of water and nutrition (He et al., 2018). However, increased antioxidant substance accumulation and increased antioxidant enzyme activities can protect plants against the oxidative injury caused by salt stress, thereby giving the plants increased resistance to salt stress (Liu et al., 2019; Wang et al., 2014). Enzymes such as superoxide dismutase in the root of Kandelia candel under varying salt stresses exhibited a trend from ascent to descent (Wang et al., 2014). The root activity of C. quinoa was significantly greater under the 150 mM NaCl treatment, compared to the control. The higher accumulation of antioxidant substances was likely due to PGPR-induced salt stress response strategy in plants. This observation suggests that the 150 mM NaCl concentration is a moderate salt treatment for C. quinoa. Significantly elevated root activity in the M30-35 treatment was also observed under 150 and 300 mM of NaCl. In a similar study, Trichoderma longibrachiatum T6 significantly increased the root activity of wheat seedlings under salt stress (Zhang et al., 2019). According to the recent research results reported by He et al. (2018), M30-35 promoted the growth of ryegrass, by enhancing root activity under salt stress. Hence, the growth promotion in C. quinoa roots might be correlated to an increase in root activity under salt stress.

Salinity interferes with the absorption of macro-elements (K, Ca, Mg, P, and S) and micro-elements (Zn, Fe, Mn, Cu, and B) by plants, leading to nutrient deficiency and metabolic disorders (Egamberdieva et al., 2017; Kim et al., 2017; Munns & Tester, 2008). P is a key nutrient element in plants and plays an important role in the process of plant growth and development. The solubilization of bound inorganic P and mobilization of organic P by PGPR are mechanisms that have a positive effect on plant growth (Liu et al., 2014; Zeng, Wu & Wen, 2016). Under high salt conditions, eggplant seedlings inoculated with Achromobacter increased the absorption of P (Mayak, Tirosh & Glick, 2004), and inoculated wheat seedlings with Bacillus increased the levels of P in seedlings (Upadhyay & Singh, 2015). Our results showed that the P content of non-inoculated C. quinoa seedlings under salt stress was significantly lower than seedlings not exposed to salt stress. This indicated that salt stress can influence P absorption. The results of the present study indicate that inoculation of M30-35 can maintain (P) homeostasis and mitigate the salt stress impact on C. quinoa.

C. quinoa is rich in saponins (belonging to the triterpenoid saponins) mainly concentrated in the pericarp (Kuljanabhagavad et al., 2008). The saponins contents vary in different C. quinoa varieties and are in the range of 47.11–136.98 mg/100g, which are often influenced by biotic and abiotic factors (Szakiel, Paczkowski & Henry, 2011). Under 200 mM of salt stress, the C. quinoa seedlings accumulated higher levels of saponins than control (Gomezcaravaca et al., 2012). Furthermore, Yang et al. (2018) demonstrated that saponins might be a biostimulant for germination in C. quinoa, especially under salt stress. Thus, saponin might be involved in the response of C. quinoa to salt stress. Our results showed that 150 mM of NaCl induced the synthesis of saponins in C. quinoa seedlings. Besides, the C. quinoa plants inoculated with M30-35 exhibited higher saponins levels compared to un-inoculated plants under 300 mM NaCl salt stress. M30-35 might enhance the salt tolerance of C.quinoa by partially increasing their saponins accumulation.

Conclusions

The effects of Pseudomonas sp. strain M30-35 on the growth and physiology of C. quinoa seedlings were evaluated. Under salt stress, M30-35 promoted the growth of C. quinoa, and increased the root and total biomass of C. quinoa seedlings. The chlorophyll a and chlorophyll b content in the 300 mM NaCl treatment was increased by inoculation of M30-35. M30-35 improved the root activity and accumulation of total saponins and alleviated the decline in P content under salinity. These data provide evidence that M30-35 inoculation of C. quinoa seedlings can improve their salt tolerance.

Supplemental Information

Supplemental Information 1 The standard curves were drawn.

From the phosphorus standards, a calibration curve was generated. The standard curve of Oleanolic acid(mg/ml) concentration was constructed from 0 to 0.7.The root activity of each group was calculated by the standard curve.

Click here for additional data file.

Supplemental Information 2 Statistical analysis was performed by one - way ANOVA

Statistical analyses of all biological data were carried out by use of one - way ANOVA. Data were analyszed using SPSS version 21.0 (IBM Corporation) statistical software.

Click here for additional data file.

Supplemental Information 3 This file contains raw source data used to make the graphs presented in Figure

The data were plotted using Origin software. Results of the growth and physiological data are presented as means ±standard error.

Click here for additional data file.

The authors acknowledge Dr. Zhao Qi for providing experimental materials. We thank LetPub for its linguistic assistance during the preparation of this manuscript.

Additional Information and Declarations

Competing Interests

Author Contributions

Data Availability

The authors declare there are no competing interests.

Deyu Cai and Ying Xu conceived and designed the experiments, performed the experiments, analyzed the data, prepared figures and/or tables, and approved the final draft.

Fei Zhao and Xiaonong Guo conceived and designed the experiments, authored or reviewed drafts of the paper, and approved the final draft.

Yan Zhang conceived and designed the experiments, performed the experiments, prepared figures and/or tables, and approved the final draft.

Huirong Duan analyzed the data, authored or reviewed drafts of the paper, and approved the final draft.

The following information was supplied regarding data availability:

Raw data are available in the Supplemental Files.

The strain M30-35 used in this study is permanently stored in the Marine Culture Collection of China with preservation number 1k03247.

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
