# Peer review of "Improved salt tolerance of Chenopodium quinoa Willd. contributed by Pseudomonas sp. strain M30-35"

_PeerJ, doi:10.7717/peerj.10702_

## Round 0.1 · original submission · Major Revisions

Dear Authors

Major revision is needed in the whole manuscript to improve its quality and make it according to the standard of PeerJ. The English is very poor and needs to be checked and improved significantly.

Amanullah

Reviewer 1 ·

Basic reporting

In my opinion, this is an acceptable manuscript for publication after corrections. In this regard, my detailed views on the current work are listed below.
- The context of the manuscript is in the scope of the journal.
- Introduction section line 32 the sentence “Add your introduction here.” should remove from the manuscript.
- The sentence in line 83 and 84 “reported that bacillus inoculum increased the plant biomass, relative water content, leaf water potential and leaf water loss, and reduced electrolyte leakage.” should have written again. Because of grammatical mistakes.
- Introduction section should have written again. Some of the sentences which are related recent examples about PGPR strain of saline soils should remove from there and please add to discussion section.
- Genus and species names should have written italic in all of the manuscript.
- There is no need for subtitles in discussion section.
- There are some grammatical mistakes in the text. It should check and revise some professional language experts or native speakers.

Experimental design

Experimental design of article is enough and all methods are generally used in this kind of studies.

Validity of the findings

The topic of article is rationale and very popular. Conclusion is are well explained. But language have to improved by authors.

Additional comments

In my opinion, this is an acceptable manuscript for publication after corrections. In this regard, my detailed views on the current work are listed below.
- The context of the manuscript is in the scope of the journal.
- Introduction section line 32 the sentence “Add your introduction here.” should remove from the manuscript.
- The sentence in line 83 and 84 “reported that bacillus inoculum increased the plant biomass, relative water content, leaf water potential and leaf water loss, and reduced electrolyte leakage.” should have written again. Because of grammatical mistakes.
- Introduction section should have written again. Some of the sentences which are related recent examples about PGPR strain of saline soils should remove from there and please add to discussion section.
- Genus and species names should have written italic in all of the manuscript.
- There is no need for subtitles in discussion section.
- There are some grammatical mistakes in the text. It should check and revise some professional language experts or native speakers.

Reviewer 2 ·

Basic reporting

General comments
The language/grammar used in the manuscript requires significant improvement. It is unacceptable in present format. Several statements lack of clarity, precision, and comprehensiveness. Moreover, the research work lacks sufficient novelty, for the reason that the effect of plant beneficial bacteria on plant growth under stress is known and many reports are available. The writing style is appalling and requires a major revision.

Experimental design

Experimental design is well organised, but english writing should be imporved.
I made some comments which could help to improve MS.

Validity of the findings

The results support the aim of work, and only limitation is that there is no novelty in this study.
I have suggested authors to write new findings and add more recent citations on related topic.

Additional comments

General comments
The language/grammar used in the manuscript requires significant improvement. It is unacceptable in present format. Several statements lack of clarity, precision, and comprehensiveness. Moreover, the research work lacks sufficient novelty, for the reason that the effect of plant beneficial bacteria on plant growth under stress is known and many reports are available. The writing style is appalling and requires a major revision.

Specific comments
Line 62: “Add your introduction here” remove
Lines 67-68: “The present two popular strategies”, which strategies, describe?
Lines 102-124: Please revise and shorten the text, many repetitions
Line 127: remove, “first time”, the beneficial effect of root associated bacteria on plant growth under abiotic stresses is known.
Lines 15-133. Please shorten text and clarify the aim of study, but do not repeat
Line 139: Luria broth, please add source
Line 146: seeds were inoculated, but why after surface-sterilized?
Line 147: germinated in plates? Or pots?
Line 172: spectrophotometer model?
Line 174: Determination of root activity? Root system architecture?
Why only P concentration in plant tissue were measured, what about N,K and other minerals?
Line 221: under both the LB medium (control) and M30-35 medium, please correct, e.g. under non saline and 100mM NaCl condition?
Line 255, please explain root activity
There are many outdated references, please add recent reports on PGPR activity on plant growth under stress. Moreover, it is important to demonstrate new findings in this study. Because it is known that Pseudomonas strains increase plant stress tolerance and also modulate physiological processes.

·

Basic reporting

no comment

Experimental design

no comment

Validity of the findings

no comment

Additional comments

In this work, Cai and co-authors described the effects of a Pseudomonas sp. strain M30-35 on the responses of quinoa plants to salt stress in attempt to explain the mechanism of the increased salt tolerance by the bacterium. The comparative assays for the physiological parameters from control and M30-35 treated plants under salinity stress indicated that M30-35 treatment could increased the salt tolerance of quinoa via improving the inhibited growth and alleviating physiological toxicity of quinoa plants exposed to salinity stress. The results are simple, but they are very worth, because the findings may be helpful to generate new bio-fertilizer which may be useful to improve the growth of crops grown in saline alkaline soil. Therefore, my overall impression is positive, but some issues should be addressed before the paper may be recommended for the publication.

Major comments
1 Tons of grammatical errors occurred throughout the manuscript, so the use of English grammar and style should be thoroughly checked and corrected. It is the best to find an expert with good English knowledge to polish your text and to avoid any grammatical and syntax errors.
For example:
Lines 103-105, “This is rich in protein, amino acids, dietary fiber, vitamins, trace elements, and various beneficial phytochemicals.”

Lines 289-292, “It might be possible that M30-35 functioned on the roots through similar mediated methods, in which KM4 improved the salt tolerance in maize by promoting plant growth to improve the salt tolerance of quinoa.”
The two sentences are only the representatives of a lot of descriptions which is very difficult to understand.

2 The authors seem to choose some physiological parameters randomly, and put them in the manuscript very simply, because the relationship among them is not close, especially saponin is rarely used in studying salt tolerant mechanism of plants. If saponin is an important component in quinoa, please describe it in detail in the revised manuscript.

Minor comments
1 Why did fresh and dry weights were used together to express the biomass in the manuscript? Normally, when the change in water content is analyzed, fresh and dry weights may appear at same time. However, the relative information is not mentioned at all. So it will be enough to only use dry weight to express biomass.

2 If the growth parameter, fresh weight, is deleted, original figure 1 and 2 should be combined into figure 1. Title of the revised figure 1 should be “The effect of M30-35 on the grow of quinoa plants under salinity stress.

3 “The columns with different letters indicate the significant differences among treatments at P<0.05.” The sentence can be seen in all figure legends. Different letters were marked for some columns, but their bars of SE overlapped. Normally, if SE bar of two data overlap, the difference between two data is not significant. So please explain for these data.

4 a lot of words are not used accurately.
For examples:
Line 208, the word “estimated” should be changed into “determined”.
Line 311, the word “subdue” should be changed into “inhibit”.

Taken together, the arrangements and English style are very poor, so please check and revise carefully the manuscript for its resubmission version.

---

## Round 0.2 · accepted · Accept

The authors incorporated all comments of the reviewers and improved the manuscript. In this manuscript impact of Pseudomonas sp. strain M30-35 on growth and physiology of C. quinoa seedlings are reported. The strain M30-35 improved growth of C. quinoa, under salinity stress (150 and 300 mmol/L).

·

Basic reporting

no comment

Experimental design

no comment

Validity of the findings

no comment

Additional comments

The authors have addressed all issues from me.